# Multiple Antibiotic Resistance (MAR), Plasmid Profiles, and DNA Polymorphisms among *Vibrio vulnificus* Isolates

**DOI:** 10.3390/antibiotics8020068

**Published:** 2019-05-28

**Authors:** Mohammed M. Kurdi Al-Dulaimi, Sahilah Abd. Mutalib, Maaruf Abd. Ghani, Nurul Aqilah Mohd. Zaini, Ahmad Azuhairi Ariffin

**Affiliations:** 1Education Supervision Department, Office of Assistant undersecretary for Educational Affairs, Ministry of Education and Higher Education, Doha, Qatar; bio_micro1978@yahoo.co.uk; 2Centre for Biotechnology and Functional Food, Faculty of Science and Technology, Universiti Kebangsaan Malaysia (UKM), Bangi 43600 UKM, Malaysia; maaruf71@ukm.edu.my (M.A.G.); nurulaqilah@ukm.edu.my (N.A.M.Z.); 3Department of Community Health, Faculty of Medicine and Health Sciences, Universiti Putra Malaysia, Serdang 43400 UPM, Malaysia; zuhairifin@upm.edu.my

**Keywords:** multiple antibiotic resistance (MAR) index, plasmid profiles, DNA polymorphisms, *Vibrio vulnificus*, genomic heterogeneity

## Abstract

Sixty strains (*n* = 60) of *Vibrio vulnificus* were examined for their multiple antibiotic resistance (MAR) index, plasmid profiles, and DNA polymorphisms. Thirty-seven strains (*n* = 37) were isolated from cockles (*Anadara granosa*) in Malaysia, while 23 (*n* = 23) isolates were isolated from clams (*Mercenaria mercenaria*) in Qatar. All isolates were resistant to two or more of the antibiotics tested, with the most common resistances were demonstrated towards penicillin (93%), ampicillin (70%), cephalothin (65%), clindamycin (66%), vancomycin (64%), and erythromycin (51%). The antibiotic that experienced the least resistance was kanamycin (6%), and all isolates were susceptible to cefoperazone, streptomycin, and tetracycline. The MAR index for the *V. vulnificus* isolated from Malaysia and Qatar, possessed similar values which ranged from 0.2 to 0.7, respectively. Plasmid analysis demonstrated that 65% of *V. vulnificus* strains harbored plasmids, while 35% were not. Nineteen (P1–P19) plasmids profiles were observed. No specific cluster or group was observed although they were isolated from different sample sources and locations by phylogenetic analysis using GelCompar II software at an 80% similarity level. Results demonstrated the high MAR index and genomic heterogeneity of *V. vulnificus*, which are of great concern to the human health of those who have consumed cockles and clams from the study area.

## 1. Introduction

*Vibrio vulnificus* is a rod-shaped, Gram-negative halophile and an opportunistic human pathogen. It belongs to the family of Vibrionaceae and is ubiquitous in marine environments. This bacterium has been isolated from water, sediments, fish, and shellfish [1,2]. *V. vulnificus* can cause diseases to individuals who eat contaminated seafood or have an open wound infected by this bacterium via seawater. Their infection can be fatal or can cause sepsis in susceptible individuals.

In the previous studies, potentially pathogenic Vibrio species such as *Vibrio parahaemolyticus*, *V. cholera*, and *V. vulnificus* were detected in seafood sold in Malaysia, the later was recently found in shrimps, squids, crabs, cockles, and mussels. The results of these studies showed that the most virulent of the non-cholera vibrios—*V. vulnificus*—has various virulence factors that facilitate the development of clinical disease [3,4]. On the other hand, no incidence of *V. vulnificus* was reported in Qatar. An investigation is greatly needed because of the current increasing concern that *V. vulnificus* may represent a clinical problem, especially in communities that consume shellfish such as cockles, oysters, and clams.

Antibiotic resistance is used as an epidemiological tool for foodborne disease control; it also provides antibiotic information, which may help to treat disease due to this bacterium. The emergence of multiple antibiotics resistance (MAR) bacteria may pose a threat to human health [5]. The emergence of MAR bacteria was due to indiscriminate use of antibiotics in clinical medicine, agriculture and aquaculture industries [6]. *Vibrio* spp. was reported to be greatly susceptible to the majority of clinically used antibiotics [7]. However, based on annual reports, an increasing number of *Vibrio* spp. have become more resistant toward clinically utilized antibiotics [8]. Antibiotics may contribute to the survival of bacteria strains that may contain resistance (R) plasmids. The transfer of R plasmids from resistant to nonresistant organisms is of great medical significance because it reduces the effective use of antibiotics. A previous study reported that there was a correlation between antibiotic resistance and the presence of the plasmids on *Vibrio* spp. [9]. Strains of biotype 2 of *V. vulnificus* possess one or more virulence plasmids [10], ranging between 68 and 70 kb. *V. vulnificus* strains were also found to carry more than one plasmid with diverse sizes [11].

Molecular approaches are useful to enable us to group bacteria into distinctive groups according to their distinctive features, between two geographical locations. Phenotype-based subtyping, such as antibiotic resistance, and deoxynucleic acid (DNA)-based subtyping, such as plasmid profiles and random amplified polymorphism DNA (RAPD) analysis, allow the bacterial isolates to be differentiated below the species level. Bacterial subtyping would help researchers to detect and track foodborne disease outbreaks, clonal species circulating in different locations, and as tools to track the sources of bacterial contamination in the food system. It also facilitates a better understanding of the ecology of different foodborne pathogens, population genetics, and epidemiology. In the present study, we investigated the antibiotic resistance, plasmid profiles and polymerase chain reaction (PCR)-based analysis of *V. vulnificus* isolated from cockles and clams from Malaysia and Qatar.

## 2. Results

### 2.1. Antibiotic Resistance

In general, a total of 60 *V. vulnificus* isolates from both countries showed resistance towards antibiotics in the following order (Table 1); penicillin (93%), ampicillin (70%), clindamycin (66%), cephlothin (65%), vancomycin (64%), bacitracin (59%), erythromycin (57%), novobiocin (46%), and kanamycin (6%). Most of the isolates in this study were sensitive to cefoperazone, streptomycin and tetracycline. Table 2 shows the antibiogram of 60 *V. vulnificus* isolates, 37 isolates from Malaysia (cockles) indicated 14 patterns (A1–A14), with the most frequent patterns being A3 (AmpKfP) and A4 (AmpBDaEKfNvPVa). While 23 *V. vulnificus* isolates from Qatar (clams) showed four new patterns (pattern A15–A18), with A1 (BDaEKfNvPVa) and A14 (AmpBDaEPVa) being the most frequently observed. By combining the two locations, 18 antibiograms were revealed (A1–A18). In the present study, Malaysia and Qatar were chosen, because to compare if there were any differences in term of multiple resistant index (MAR), plasmid profiles and DNA heterogeneities between two distantly geographical locations of *V. vulnificus* isolates.

### 2.2. Multiple Antibiotic Resistance (MAR) Index

*V. vulnificus* isolates from Malaysia and Qatar had different multiple antibiotic resistance (MAR) indexes, ranging from 0.2 to 0.7, respectively. Coincidently, both *V. vulnificus* isolates from Malaysia and Qatar had similar MAR values, ranging from 0.2 to 0.7, respectively (Table 3). Table 3 details the percentage of occurrence of each MAR index of *V. vulnificus* isolates from cockle samples from Malaysia and clam samples from Qatar. Although they have a similar MAR index, the percentage of occurrence in both places was different. A total of 15 *V. vulnificus* isolates (25%) were resistant to three antibiotics, 14 (23%) to six antibiotics, 12 (20%) to seven antibiotics, eight (13%) to eight, five (8%) to two antibiotics, and three (5%) to both four and five antibiotics, respectively (Table 2).

### 2.3. Plasmid Profiles

Among the isolated *V. vulnificus* (*n* = 60) from Malaysia and Qatar, only 40 (67%) of the strains harbored plasmids, while the other 20 *V. vulnificus* isolates (33%) did not contain any plasmid (Table 4). Since the Lambda DNA-HindIII DNA ladder is the linear DNA, the determination of plasmids’ molecular weight was based on the plasmid profiling. Plasmid profiles from cockle samples from Malaysia were denoted as P1–P11, while, P3, P6, and P12–P19 represented plasmid profiles from clam samples from Qatar (Table 2). Overall, 19 different plasmid profiles were observed as indicated in Appendix A. Figure 1, Figure 2 and Figure 3 gives examples of plasmid profiles for several strains of *V. vulnificus*.

### 2.4. RAPD-PCR

In RAPD analysis, using primer RAPD 11 and RAPD 15, the 60 *V. vulnificus* isolates generated 47 and 50 RAPD patterns, respectively. Using primer RAPD 11, 27 RAPD patterns were produced by *V. vulnificus* isolates from Malaysia and are denoted as C1 to C27. From Qatar, 20 RAPD patterns, denoted as C28–C47, were observed (Table 2). Analysis using gel compare II software differentiated the *V. vulnificus* into 15 clusters and 16 isolates at an 80% similarity level (Figure 4). Primer RAPD15 produced 32 RAPD patterns from *V. vulnificus* isolates from Malaysia, while 18 patterns were produced by *V. vulnificus* isolates from Qatar. Using gel compare II analysis, six clusters and three single isolates at the same level were analyzed as shown in Figure 5.

## 3. Discussion

The main objective of this study is to determine the antibiotic resistance, plasmid profile, and RAPD analysis of *V. vulnificus* isolated from cockles (*Anadara granosa*, Malaysia) and clams (*Mercenaria mercenaria*, Qatar). As indicated in Table 1, all isolates were tested against several antibiotics and the highest antibiotic resistance was observed towards penicillin (93%), followed by ampicillin (70%), clindamycin (66%), cephalotin (65%), vancomycin (64%), bacitracin (56%), erythromycin (51%), novobiocin (46%), and kanamycin (6%). None of these isolates were found to be resistant to cefoperazone, streptomycin, or tetracycline (Table 2). The isolates from cockle samples from Malaysia demonstrated marginally more resistance to many antibiotics compared to clam samples from Qatar, except for ampicillin and vancomycin. Thus, differences that were found in the antibiotic resistance may depend on the sample source.

The highest percentage of resistance towards penicillin (93%) was observed in the present study. These findings were consistent with the previous work, where the penicillin-resistant vibrio has been reported to be 100% resistance towards penicillin in India leading to concern regarding drug-resistant microbial diseases in aquaculture [6]. Two separate studies by Radu et al. [12,13] found that *V. vulnificus* is highly resistant to both bacitracin and penicillin in Malaysia, which was in agreement regarding penicillin but in contrast with what was previously known regarding bacitracin. In this study, only 56% of *V. vulnificus* isolates showed resistance towards bacitracin. Furthermore, ~70% of the isolates were resistant to ampicillin, the second highest resistance level after penicillin. Ampicillin was detected as having a high resistance against *V. vulnificus* isolated from the Arabian Gulf compared to the other antibiotics tested [11], which is in agreement with this study. The high susceptibility of all isolates against tetracycline and streptomycin in the present study indicated that the *V. vulnificus* was sensitive to those antibiotics compared to other Vibrio species. Son et al. [13] and Okoh et al. [14] reported some of their vibrio isolates were resistant toward streptomycin and tetracycline.

In this study, the *V. vulnificus* isolates showed high incidences of antibiotic resistance against more than two or more antibiotics. The multiple antibiotic resistance (MAR) index from cockle samples from Malaysia and clam samples from Qatar showed different MAR index values in the range of 0.2 to 0.7 (Table 3). Most of the isolates proved to be resistant to multiple antibiotics (MAR). Fifteen isolates (25%) were resistant to three antibiotics, 14 (23%) to six antibiotics, 12 (20%) to seven antibiotics, eight (13%) to eight antibiotics, five (8%) to two antibiotics, and three (5%) to both four and five antibiotics, respectively. These findings were similar to those reported by Roig and Amaro, Baker-Austin et al., Tunung et al., and Lee et al. [10,15,16,17], in which it was observed that *V. vulnificus* isolates were resistant to two or more antibiotics with a high MAR index. A MAR index value higher than 0.2 is said to have originated from high-risk sources of antibiotic contamination where antibiotics are often used, such as from human, commercial poultry farms, swine and dairy cattle [17,18]. The occurrence of antibiotic-resistant *V. vulnificus* in seafood represents a potential hazard to human health, especially to people who consume seafood that has been improperly prepared.

As indicated in Table 3, comparing between Malaysia and Qatar, although the MAR index from both places was similar (0.2–0.7), the occurrence percentages were different. For example, more *V. vulnificus* isolates from cockles samples from Malaysia had a MAR index value of 0.7 (19%) compared to clams samples from Qatar (4%). The MAR value index value of 0.7 for both exhibited resistances to eight of the antibiotics tested (AmpBDaEKfNvPVa); however, the number of isolates from cockle samples from Malaysia was almost five times higher compared to clam samples from Qatar. It has been suggested that a high number of isolates from cockle samples from Malaysia may come from the continued agricultural use of medicated feeds in animal husbandry which disseminate the virulent and resistant bacterial pathogens through the feces, resulting in dispersal into the environment. It is possible that the plasmid exchanged between bacteria in aquatic systems would also contribute to the high frequency of MAR incidences [11,14].

Plasmid is one of the most important mediators that facilitate the fast spread of antibiotic resistance among bacteria [18]. The transferal of genetic elements of antibiotic resistance to other bacteria can cause illness in humans [19]. When the resistance of isolates carrying plasmids was compared with that of isolates without plasmids, the results were almost similar for resistance towards bacitracin, clindamycin, erythromycin, and kanamycin. However, more isolates containing plasmids were resistant to ampicillin, cephalotin, novobiocin, penicillin, and vancomycin. As indicated in Table 4, 67% (40/60) of *V. vulnificus* isolates harbored plasmids, while 23% (20) of isolates did not. Approximately 26 of 37 (70%) of isolates from cockle samples and 14 from 23 (60%) isolates from clam samples harbored plasmid DNAs. This finding suggested that the resistance of *V. vulnificus* isolates was encoded on plasmids for ampicillin, cephalotin, novobiocin, penicillin, and vancomycin, or on chromosomes for bacitracin, clindamycin, erythromycin, and kanamycin. Aoki et al. [20] reported that the antibiotic resistance of *V. anguillarum* occurred in R plasmid and genome DNA. Furthermore, the antibiotic resistance of nalidixic acid and furazolidone were not transferred to *Escherichia coli*, indicating that they were present in genome DNA. However, for chloramphenicol, sulfonamides, streptomycin, ampicillin, and trimethoprim, the transferable R plasmids were carried by the strains. However, no conclusion can be drawn since the conjugation analysis has not been conducted.

The plasmid analysis of all the isolates (Table 2 and Table 4), provide a general picture of plasmids in *V. vulnificus* isolates. Of the 60 strains analyzed, 40 (67%) of the *V. vulnificus* strains harbored plasmids and 19 different plasmid profiles were observed. The most frequent plasmid profiles were P3 and P2. The multiple plasmids harbored in *V. vulnificus* isolates are in agreement with ElHadi [11] and [12], who reported the occurrence of multiple plasmids in *V. vulnificus* isolates. A total of 19 plasmid profiles were shown to be heterogeneous, suggesting it is useful as a tool for categorizing typing *V. vulnificus* isolates. The variations in the plasmid size of *V. vulnificus* strains found in this study support the findings of Zhang et al. and Zhang et al. [21,22] who observed multiple plasmids in Vibrio with variations in size. In the present study, plasmid profiles from cockle samples showed 11 profiles (P1–P11), while plasmid profiles from clam samples were mostly P12 to P19. There were two plasmid profiles from Qatar that were similar to Malaysian plasmid profiles, P3 and P6, which suggested they may have similar molecular weights but may be different in terms of their sequences.

The total number of plasmids in any given bacterial population can affect the results of the isolate analysis. For example, isolates 10 and 11 and isolates 51 and 52. These isolates possessed similar antibiotic resistances; however, their plasmid profiles differed. Without plasmid profiles, we may have believed that the *V. vulvificus* isolates with similar antibiotic resistances originated from the same ancestral isolates. Eighteen (18) antibiogram and 19 plasmid profiles were revealed in the present study, and by the combination of both methods, 43 strain types were observed. However, when these isolates were analyzed using RAPD analysis, the degree of differentiation among the strains increased, resulting in 60 strain types. The results were as expected and in agreement with [4,9,23,24] who reported the used of the RAPD technique in strains differentiation of *Escherichia coli*, *Bacillus cereus*, and *Vibrio parahaemolyticus*, respectively. RADP analysis showed 47 and 50 RAPD profiles using primer RAPD11 and primer RAPD 15, respectively (Table 3, Figure 4 and Figure 5). It was also found that the RAPD profiles from cockles from Malaysia and clams from Qatar were different, but did not show any specific cluster that differentiated between the two distant geographical locations. Two isolates (strain 28 and 30) were untypable using primer RAPD11, and a single isolate was untypable (Isolate 50) using RAPD15 primer, respectively. These results were probably due to the loss of a specific site in their genome DNA.

## 4. Materials and Methods

### 4.1. Vibrio vulnificus

A total of 60 *V. vulnificus* strains were stored in the Laboratory of Food Sciences, Universiti Kebangsaan Malaysia, Selangor. These strains were previously isolated from two distant countries, Malaysia and Qatar. Thirty-seven were from Malaysia, namely, *V. vulnificus* strains 1–37, while strains 38–60 were from Qatar (23 strains). All strains were isolated between July 2013 and February 2014.

### 4.2. Antibiotic Resistance

Antibiotic susceptibility tests were performed by the disc diffusion method on Muller Hinton agar (MHA) (Oxoid, UK) as described by Bauer et al. [25]; antibiotic discs are listed in Table 1. A single colony was cultured in 10 mL of Alkaline Peptone Water (APW) (Oxoid) and incubated overnight at 37 °C. The solution was evenly distributed over the MHA using a sterile cotton bud and was allowed to dry for 2 to 5 min. Antibiotic discs were fixed onto the agar plates by using sterilized forceps and were incubated at 37 °C for 24 h. The clear zone for each antibiotic disc was determined by measuring the diameter of the inhibition zone around the antibiotic disc.

### 4.3. Multiple Antibiotic Resistance (MAR) Index

The MAR index of the isolates against the tested antibiotics was calculated based on the following formula [17]. MAR index (multiple antibiotic resistance) = X/(Y × Z); where X = total number of antibiotic resistance cases; Y = total number of antibiotics used in the study; and Z = total number of bacterial isolates. A MAR index value of equal or less than 0.2 was defined as those antibiotics that were rarely or never used for the animal in terms of treatment; however, if the MAR index value was higher than 0.2, this was considered as an indicator of the high risk of exposure to those antibiotics received by the animals.

### 4.4. Plasmid Analysis

*V. vulnificus* strains were grown overnight in 10 mL of Lauria Bertani (LB) broth with an addition of 3% (*w*/*v*) of sodium chloride (NaCl) at 37 °C with shaking 200 rpm. A quantity of 1 mL of culture was then transferred into a 1.5 mL centrifuge tube and spun for 1 min at 10,000 rpm via a benchtop centrifuge (Minispin, Eppendorf, Germany). Plasmid extractions were conducted using PureYield™ Plasmid Miniprep System (Promega, Madison, WI, USA) as described in the manufacturer’s instructions. DNA plasmid was then analyzed on 1% (*w*/*v*) agarose gel. The gel was electrophoresed at 85 V for 1 h and Lambda DNA-HindIII Digested DNA was used as a DNA ladder (New England BioLabs, Ipswich, MA, USA). The gel was visualized using gel documentation (Syngene, Frederick, MD, USA).

### 4.5. RAPD Fingerprinting

A random amplified polymorphism DNA-polymerase chain reaction (RAPD-PCR) was used for characterization of the *V. vulnificus* isolated isolates. The RAPD11 primer used was 5′-AAAGCTGCGG-3′ and RAPD15 5′-CACACTCCAG-3′. The PCR technique was carried out in 0.2-μL microfuge tubes. The total volume of the reaction mixture was 50 μL, consisting of 25 μL 10× PCR master mix (EconoTaq^®^PULS GREEN 2X Master Mix, Lucigen, Middleton, WI, USA), 0.5 μL of OPC primer, and 1.0 μL (10–20 ng) of template DNA, the volume was then adjusted to a final volume by adding Nuclease Free Water (NFW). Concerning the negative control, one of the reaction mixtures without the DNA template was used. The solution mixture was placed in the Thermal Cycler (Bio-Rad, Hercules, CA, USA) and the PCR cycles parameters were denatured at 94 °C for 5 min followed by 45 cycles of denaturation at 94 °C for 1 min, annealing at 34 °C for 1 min, and polymerization at 72 °C for 2 min. Final elongation was carried out at 72 °C for 7 min. The 1 kb DNA ladder (Vivantis, Selangor, Malaysia) was used as a DNA size marker and fragments were viewed using a UV transilluminator (Syngene, Cambridge, UK).

### 4.6. Phylogenetic Analysis

Clonal relatedness of the *V. vulnificus* using primers RAPD11 and RAPD15 were analyzed as described by Sahilah et al. [26]. The image of the gel was analyzed using Gel ComparII (Applied Math, Kortjik, Belgium).

## 5. Conclusions

In conclusion, a high MAR index of *V. vulnificus* isolates in the study area indicated that the isolates originated from high-risk sources of contamination. No conclusion can be drawn as to the existence of R plasmids since no conjugation analysis was performed. However, in the present study, there was evidence that the resistant gene for certain antibiotics may be positioned either in plasmids or in genome DNA. While subtyping would increase the degree of strains differentiation and molecular approaches of RAPD, it is useful to differentiate *V. vulnificus* isolates, and thus increase the heterogeneity level of these isolates.

## 6. Patents

No patents have resulted from the work reported in this manuscript.

## Figures and Tables

**Figure 1 antibiotics-08-00068-f001:**
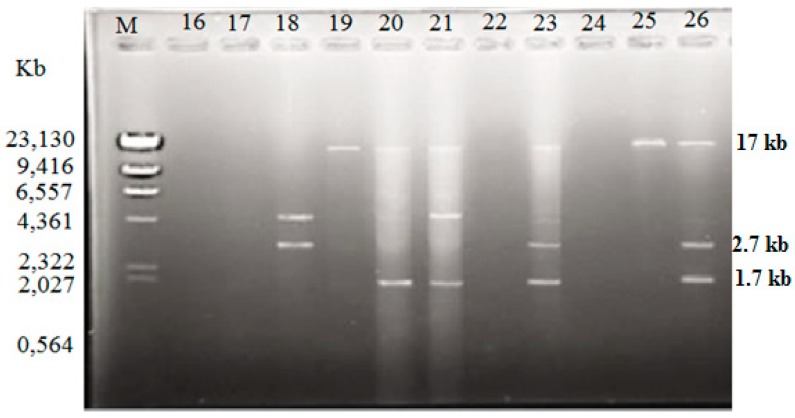
Example of plasmid profiles of *Vibrio vulnificus* isolated from Malaysia (strain 16–26) using a PureYield™ Plasmid Miniprep System on 1% (*w*/*v*) agarose gel. Lane M: Lambda DNA-*Hind*III DNA ladder. Lane 16–26: *V. vulnificus* strain 16–26.

**Figure 2 antibiotics-08-00068-f002:**
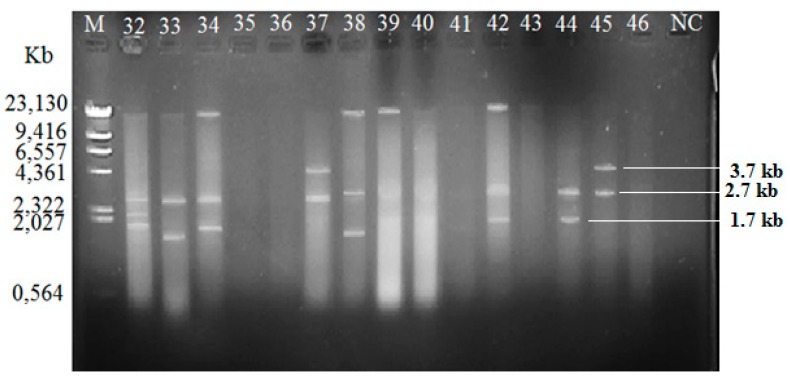
Example of plasmid profiles of *Vibrio vulnificus* isolated from Qatar (strain 32–37, Malaysia; strain 38–46, Qatar) using a PureYield™ Plasmid Miniprep System on 1% (*w*/*v*) agarose gel. Lane M: Lambda DNA-HindIII Digest DNA ladder. Lane 47–60: *V. vulnificus* strain 47–60; NC: negative control.

**Figure 3 antibiotics-08-00068-f003:**
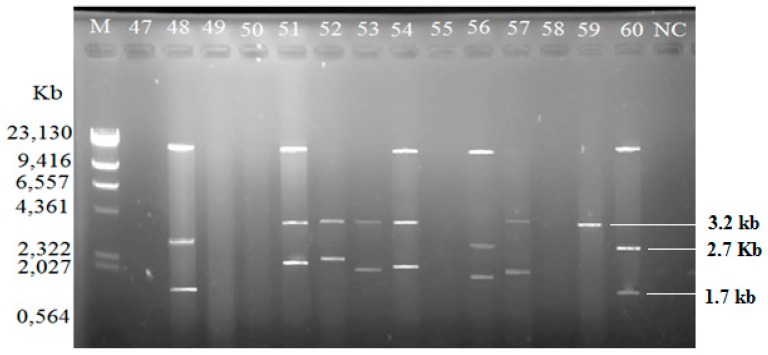
Example of plasmid profiles of *Vibrio vulnificus* isolated from Qatar (strain 47–60, Qatar) using a PureYield™ Plasmid Miniprep System on 1% (*w*/*v*) agarose gel. Lane M: Lambda DNA-HindIII Digest DNA ladder. Lane 47–60: *V. vulnificus* strain 47–60; NC: negative control.

**Figure 4 antibiotics-08-00068-f004:**
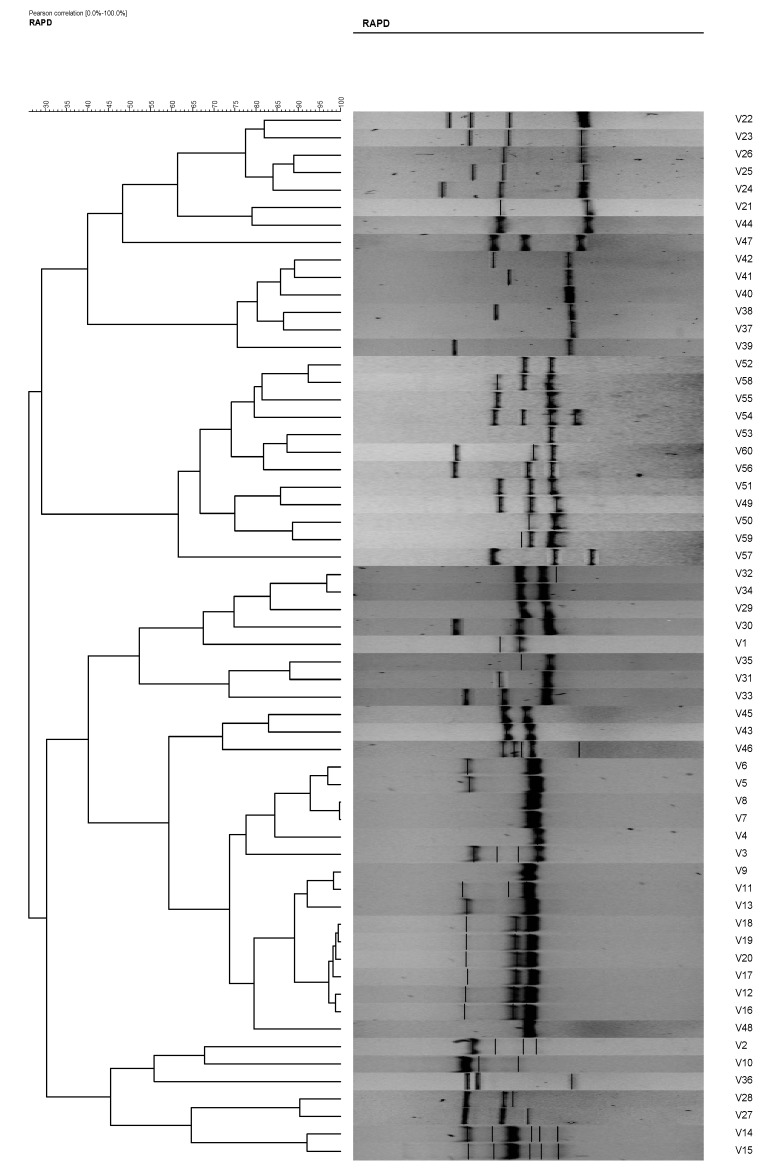
Phylogenetic tree of *Vibrio vulnificus* isolates (1–60) from RAPD analysis using primer RAPD11, which is able to differentiate the *V. vulnificus* into 15 clusters and 16 isolates at an 80% similarity level.

**Figure 5 antibiotics-08-00068-f005:**
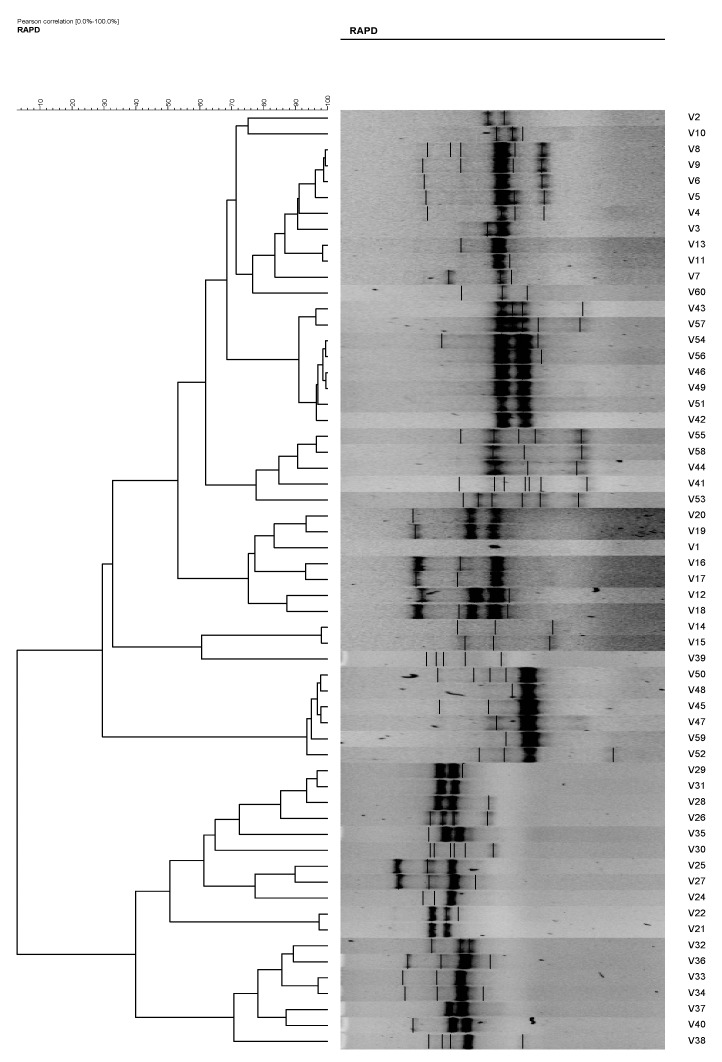
Phylogenetic tree of *Vibrio vulnificus* isolates (1–60) from RAPD analysis using primer RAPD15, which is able to differentiate into six clusters and three single isolates at an 80% similarity level.

**Table 1 antibiotics-08-00068-t001:** Frequency of antibiotic resistance *Vibrio vulnificus* isolates from cockle and clam samples.

Antibiotic Tested	Percentage of Resistance (%)		
	Cockle Samples, Malaysia (*n* = 37)	Clam Samples, Qatar (*n* = 23)	Mean Average
Ampicillin	65	74	70
Bacitracin	59	52	56
Cefoperazone	0	0	0
Cephalothin	73	57	65
Clindamycin	70	61	66
Erythromycin	57	45	51
Kanamycin	0	13	6
Novobiocin	57	35	46
Penicillin	95	91	93
Streptomycin	0	0	0
Tetracycline	0	0	0
Vancomycin	62	65	64

**Table 2 antibiotics-08-00068-t002:** Multiple antibiotic resistance (MAR), antibiotic resistance, plasmid profiling, and random amplified polymorphic DNA (RAPD) analysis of *Vibrio vulnificus* isolated from cockles (*Anadara granosa*) from Malaysia and clams (*Mercenaria mercenaria*) from Qatar.

Strain No. ^a^	Locations	MAR Index	Antibiotic Patterns ^b^		Plasmid Profiles ^c^	RAPD		Strain Types ^d^
						RAPD 11	RAPD 15	
**1**	Malaysia	0.53	BDaEKfNvPVa	A1	P1	C1	F1	1
**2**	Malaysia	0.53	BDaEKfNvPVa	A1	P2	C1	F1	2
**3**	Malaysia	0.5	AmpDaEKfNvP	A2	P2	C2	F2	3
**4**	Malaysia	0.53	BDaEKfNvPVa	A1	P2	C3	F3	4
**5**	Malaysia	0.25	AmpKfP	A3	P2	C4	F4	5
**6**	Malaysia	0.66	AmpBDaEKfNvPVa	A4	P3	C4	F5	6
**7**	Malaysia	0.5	BDaKfNvPVa	A5	P4	C5	F6	7
**8**	Malaysia	0.5	BDaKfNvPVa	A5	-	C6	F7	8
**9**	Malaysia	0.53	AmpBDaKfNvPVa	A6	-	C6	F8	9
**10**	Malaysia	0.25	AmpKfP	A3	P5	C7	F9	10
**11**	Malaysia	0.35	AmpKfP	A3	P3	C8	F10	11
**12**	Malaysia	0.16	PVa	A7	P1	C8	F11	12
**13**	Malaysia	0.66	AmpBDaEKfNvPVa	A4	P6	C9	F12	13
**14**	Malaysia	0.66	AmpBDaEKfNvPVa	A4	-	C10	F13	14
**15**	Malaysia	0.5	AmpDaEKfNvP	A2	-	C11	F14	15
**16**	Malaysia	0.25	AmpKfP	A3	-	C12	F15	16
**17**	Malaysia	0.25	AmpKfP	A3	-	C13	F16	17
**18**	Malaysia	0.58	AmpBDaKfNvPVa	A6	P7	C14	F17	18
**19**	Malaysia	0.66	AmpBDaEKfNvPVa	A4	P8	C15	F18	19
**20**	Malaysia	0.25	KfPVa	A8	P9	C15	F19	20
**21**	Malaysia	0.58	AmpBDaKfNvPVa	A6	P10	C16	F20	21
**22**	Malaysia	0.66	AmpBDaEKfNvPVa	A4	-	C16	F20	22
**23**	Malaysia	0.66	AmpBDaEKfNvPVa	A4	P2	C16	F21	23
**24**	Malaysia	0.25	AmpPVa	A12	-	C16	F21	24
**25**	Malaysia	0.25	AmKfP	A3	P8	C17	F22	25
**26**	Malaysia	0.33	BDaEP	A9	P2	C18	F23	26
**27**	Malaysia	0.5	BDaENvPVa	A10	P3	C19	F24	27
**28**	Malaysia	0.33	AmpBDaE	A11	-	UT	F25	28
**29**	Malaysia	0.66	AmpBDaEKfNvPVa	A4	P3	C20	F25	29
**30**	Malaysia	0.33	BDaEP	A9	P3	UT	F26	30
**31**	Malaysia	0.25	AmpPVa	A12	P1	C21	F27	31
**32**	Malaysia	0.5	AmpDaEKfNvP	A2	P11	C22	F28	32
**33**	Malaysia	0.5	BDaENvPVa	A10	P3	C23	F29	33
**34**	Malaysia	0.16	KfVa	A13	P3	C24	F29	34
**35**	Malaysia	0.5	AmpBDaEPVa	A14	-	C25	F30	35
**36**	Malaysia	0.33	BDaEP	A9	-	C26	F31	36
**37**	Malaysia	0.5	AmpDaEKfNvP	A2	P7	C27	F32	37
**38**	Qatar	0.66	AmpBDaEKKfNvP	A15	P12	C28	F33	38
**39**	Qatar	0.41	AmpKNvPVa	A16	P13	C28	F34	39
**40**	Qatar	0.5	AmpDaEKfNvP	A2	-	C29	F35	40
**41**	Qatar	0.41	AmpDaKPVa	A17	-	C30	F36	41
**42**	Qatar	0.25	AmpPVa	A12	P14	C31	F37	42
**43**	Qatar	0.5	AmpBDaEPVa	A14	-	C32	F38	43
**44**	Qatar	0.25	AmpPVa	A12	P3	C34	F40	44
**45**	Qatar	0.5	AmpBDaEPVa	A14	P6	C35	F41	45
**46**	Qatar	0.25	AmpKfP	A3	-	C16	F21	46
**47**	Qatar	0.58	AmpBDaKfNvPVa	A6	-	C36	F42	47
**48**	Qatar	0.41	AmpBDaEP	A18	P15	C36	F43	48
**49**	Qatar	0.25	AmpKfP	A3	-	C37	F43	49
**50**	Qatar	0.5	AmpBDaEPVa	A14	-	C38	F44	50
**51**	Qatar	0.58	BDaEKfNvPVa	A1	P16	C39	F45	51
**52**	Qatar	0.58	BDaEKfNvPVa	A1	P17	C40	F46	52
**53**	Qatar	0.58	BDaEKfNvPVa	A1	P18	C41	F47	53
**54**	Qatar	0.5	AmpBDaEPVa	A14	P16	C42	F48	54
**55**	Qatar	0.5	AmpBDaEPVa	A14	-	C43	UT	55
**56**	Qatar	0.25	AmpKfP	A3	P17	C44	F49	56
**57**	Qatar	0.25	AmpKfP	A3	P18	C45	F50	57
**58**	Qatar	0.16	KfVa	A13	-	ND	F50	58
**59**	Qatar	0.16	KfVa	A13	P19	C46	F50	59
**60**	Qatar	0.58	BDaEKfNvPVa	A1	P15	C47	F50	60

^a^*V. vulnificus* isolates **1**–**37** isolated from Malaysia, **38**–**60** isolates from Qatar. ^b^ Tested for Ampicillin (Amp), Bacitracin (B), Cephalothin (Kf), Clindamycin (Da), Erythromycin (E), Kanamycin(K), Novobiocin (Nv), Penicillin (P), Streptomycin (S), and Vancomycin (Va). ^c^ Tested for plasmid profiles. ^d^ Combination between antibiotic resistance, plasmid profiling, and random amplified polymorphism DNA (RAPD) fingerprinting. A1–A18, antibiotic patterns, and P1–P21, plasmid patterns. MAR—multiple antibiotics resistance; ND—not determined; and UT—untypable.

**Table 3 antibiotics-08-00068-t003:** Multiple antibiotic resistance (MAR) index of *Vibrio vulnificus* isolates (*n* = 60) from Malaysia (cockles) and Qatar (clams).

MAR Index	Percentage of Occurrence (%)	
	Isolates from Malaysia (Cockles) (*n* = 37)	Isolates from Qatar (Clams) (*n* = 23)
0.1	-	-
0.2	22	32
0.3	16	-
0.4	3	13
0.5	35	30
0.6	5	17
0.7	19	4
0.8	-	-
0.9	-	-
1.0	-	-

**Table 4 antibiotics-08-00068-t004:** Antibiotic resistance and plasmid occurrence of *Vibrio vulnificus* isolates from cockle and clam samples purchased from wet markets in Malaysia and Qatar, respectively.

Antibiotic Tested	Mean Averages (%) Isolates Resistant within Two Sample Sources	No. of *V. vulnificus* Isolates Resistant	
		With Plasmid (40)	Without Plasmid (20)
Ampicillin	70	36 (90%)	15 (75%)
Bacitracin	56	23 (58%)	11 (55%)
Cefoperazone	0	0	0
Cephalothin	65	28 (70%)	12 (60%)
Clindamycin	66	27 (68%)	13 (65%)
Erythromycin	51	22 (55%)	9 (45%)
Kanamycin	6	2 (5%)	1 (5%)
Novobiocin	46	23 (58%)	6 (30%)
Penicillin	93	39 (98%)	17 (85%)
Streptomycin	0	0	0
Tetracycline	0	0	0
Vancomycin	64	27 (68%)	11 (55%)

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
