# Peer review of "Multiple Antibiotic Resistance (MAR), Plasmid Profiles, and DNA Polymorphisms among Vibrio vulnificus Isolates"

_antibiotics, 2019, doi:10.3390/antibiotics8020068_

Round 1
Reviewer 1 Report
The manuscript: “Multiple antibiotic resistance (MAR), plasmid profiles and DNA polymorphisms among Vibrio vulnificus isolates” is well written and data are nicely presented. However, there are some minor grammatical errors so a careful read through prior to submission would help.
Author Response
Response to Reviewer 1 Comments
Point 1: “Multiple antibiotic resistance (MAR), plasmid profiles and DNA polymorphisms among Vibrio vulnificus isolates” is well written and data are nicely presented. However, there are some minor grammatical errors so a careful read through prior to submission would help.
Response 1: English language has been edited using MDPI English services.

Reviewer 2 Report
Significant English language editing is required for consideration to be published.
The plasmid analysis is inconclusive to the nature and significance of the presence/absence of plasmids. Size differences on the agarose gels could be from fragmented chromosome, similar plasmids with insertion elements, or a number of other possibilities.
Without sequence verification of the plasmids, it is pure speculation as to their possession of AR genes or not.
Major revisions should be completed to improve the soundness of the study.
Author Response
Response to Reviewer 2 Comments
Point 1: Significant English language editing is required for consideration to be published.
Response 1: English language has been edited using MDPI English services.
Point 2: The plasmid analysis is inconclusive to the nature and significance of the presence/absence of plasmids. Size differences on the agarose gels could be from fragmented chromosome, similar plasmids with insertion elements, or a number of other possibilities.
Response 2: Plasmid analysis is part of our study to characterize the V. vulnificus isolates according to standard protocol of plasmid extraction using PureYield™ Plasmid Miniprep System (Promega, USA). Thus, we are certain with the protocol that the fragmented chromosome is not available.
Point 3: Without sequence verification of the plasmids, it is pure speculation as to their possession of AR genes or not.
Response 3: It is impossible to do sequence verification of the plasmids due no database sequence for V. vulnificus plasmid.
Point 4: Major revisions should be completed to improve the soundness of the study.
Response 4: The English language has been improved and this will give better understanding of the manuscript.

Reviewer 3 Report
1) The order of the content has to be changed. Methods section should be between introduction and results.
2) Similarly, table 2 should follow table 1 and not table 3.
3) What is the rationale for choosing Qatar and Malaysia and not other countries? Please explain in the last paragraph of introduction.
4) The authors have represented the A1-A18 antibiogram patterns as a string of abbreviations of the antibiotics names that the bacteria are not susceptible to i.e, the strains are resistant to these antibiotics. Correct? If yes, is this a universally accepted method of representing antibiogram patterns?
5) What do the numbers in the supplementary figure represent? diameter of the inhibition zone around the antibiotic disc? If yes, please include units.
6) In the formula for MAR index calculation, cases in the numerator means patients? I see multiple formulas for calculating MAR index, but this one is not that common. Kindly cite the reference you have followed for the calculation used in this study.
7) The authors have this statement in the conclusion: 'But in the present study, there were evident that the resistance gene for certain antibiotic may position either on plasmids or on genome DNA.' I cannot find the evidence for this in the data. Please explain.
Author Response
Response to Reviewer 3 Comments
Point 1: The order of the content has to be changed. Methods section should be between introduction and results.
Response 1: The order of the content cannot be changed due to the journal requirement.
Point 2: Similarly, table 2 should follow table 1 and not table 3.
Response 2: The amendment has been done according to the table order. The paragraph has been shifted to Line 126 -138.
Point 3: What is the rationale for choosing Qatar and Malaysia and not other countries? Please explain in the last paragraph of introduction.
Response 3: The rationale for choosing Qatar is:
i. To compare if there are any differences in term of MAR, plasmids and DNA heterogeneities between two distantly geographical locations of V. vulnificus isolates
ii. Since the PhD candidate is come from Qatar we decided to compare the V. vulnificus isolates between two countries.
Point 4: The authors have represented the A1-A18 antibiogram patterns as a string of abbreviations of the antibiotics names that the bacteria are not susceptible to i.e, the strains are resistant to these antibiotics. Correct? If yes, is this a universally accepted method of representing antibiogram patterns?
Response 4: Yes, this is the normally way to present the antibiotic resistance isolates. Antibiogram is the pattern of the antibiotic resistance.
Point 5: What do the numbers in the supplementary figure represent? diameter of the inhibition zone around the antibiotic disc? If yes, please include units.
Response 5: The antibiotic resistances were based on the inhibition zone around the antibiotic disk. It has been mentioned in the supplementary table, the inhibition unit was in cm (the cm is stated at table below).
Point 6: In the formula for MAR index calculation, cases in the numerator means patients? I see multiple formulas for calculating MAR index, but this one is not that common. Kindly cite the reference you have followed for the calculation used in this study.
Response 6: The MAR index used in this study as described by Lee, S.W.; Wendy, W. Antibiogram and Heavy Metal Resistance Pattern of Salmonella spp. Isolated from Wild Asian Sea Bass (Lates calcarifer) from Tok Bali, Kelantan, Malaysia. Jor. J. Biol. Sci., 2011, 4(3), 125-128.
Reference no. 26 in the manuscript.
Point 7: The authors have this statement in the conclusion: 'But in the present study, there were evident that the resistance gene for certain antibiotic may position either on plasmids or on genome DNA.' I cannot find the evidence for this in the data. Please explain.
Response 7: The data in Table 4, supported the statement. For example ampicillin, 90% were indicated resistant on plasmid (with plasmid) whereas, 75% (without plasmid). While, penicillin, 98% were indicated resistant on plasmid (with plasmid), while 85% (without plasmid).
All antibiotic resistances were almost having comparable value within with plasmid and without plasmid. Thus, the resistance gene for certain antibiotic may position either in plasmids or in genome DNA.

Round 2
Reviewer 1 Report
I am satisfied with the current form of the manuscript.
Author Response
Response to Reviewer 1 Comments
Point 1: I am satisfied with the current form of the manuscript.
Response 1: Thank you very much.

Reviewer 2 Report
The language was significantly improved.
This reviewer still has some concerns about the suggestion of AR genes on plasmids with no genetic evidence to support those claims. It would be possible to send the plasmids for sequencing, and once the contigs were assembled, the sequence could be BLAST'ed against databases for AR genes. The full sequence of the plasmid is not the concern; it is whether it contains AR genes or not.
Author Response
Response to Reviewer 2 Comments
Point 1: The language was significantly improved.
Response 1: Agreed.
Point 2: This reviewer still has some concerns about the suggestion of AR genes on plasmids with no genetic evidence to support those claims. It would be possible to send the plasmids for sequencing, and once the contigs were assembled, the sequence could be BLAST'ed against databases for AR genes. The full sequence of the plasmid is not the concern; it is whether it contains AR genes or not.
Response 2: Some concerns about the suggestion of AR genes on plasmids with no genetic evidence to support those claims-
In general, to determine the possible occurrence of AR gene whether on plasmid or chromosome was done using simple analysis as shown in Table 4. In this analysis the existence of plasmid and without plasmid was done against 12 antibiotics tested. The other possible experiment to determine AR gene is on plasmid by doing conjugation analysis, however we didn’t do the analysis, thus our conclusion as stated in Line 281-282; “However, no conclusion can be drawn since the conjugation analysis has not been conducted”.
Example in Ampicillin, novobiocin, penicillin and vancomycin.
(Line 319-320) This finding, suggested that the resistance of V. vulnificus isolates was encoded on plasmids for ampicillin, cephalotin, novobiocin, penicillin and vancomycin; this statement was tight with “However, no conclusion can be drawn since the conjugation analysis has not been conducted”. Using conjugation analysis the determination of AR genes are transferable (F plasmid) or not by selection of transconjugants on medium added with antibiotics. Thus, our experiment is not able to answer the AR genes are on plasmid (yes or no). Our experiments use the antibiogram and plasmid profiles to characterized the Vibrio vulnificus between two distantly locations. We feel difficult to sequence the 19 plasmid profiles due to it high cost and time consuming though the full sequences are not needed.
Table 4. Antibiotic resistant and plasmid occurrence of Vibrio vulnificus isolates from cockle and clam samples purchased from wet market in Malaysia and Qatar, respectively.
| |||
Antibiotic tested | Mean averages (%) isolates resistant within two sample sources | No. of V. vulnificus isolates resistant | |
With plasmid (40) | Without plasmid (20) | ||
Ampicillin | 70 | 36 (90 %) | 15 (75%) |
Bacitracin | 56 | 23 (58 %) | 11 (55 %) |
Cefoperazone | 0 | 0 | 0 |
Cephalothin | 65 | 28 (70 %) | 12 (60%) |
Clindamycin | 66 | 27 (68 %) | 13 (65%) |
Erythromycin | 51 | 22 (55 %) | 9 (45 %) |
Kanamycin | 6 | 2 (5%) | 1 (5 %) |
Novobiocin | 46 | 23 (58%) | 6 (30%) |
Penicillin | 93 | 39 (98%) | 17 (85%) |
Streptomycin | 0 | 0 | 0 |
Tetracycline | 0 | 0 | 0 |
Vancomycin | 64 | 27 (68 %) | 11 (55 %) |

Reviewer 3 Report
1) Please include the explanation for choosing Qatar under results.
2) English language still needs improvement. Please use a software to ensure correct usage.
Author Response
Response to Reviewer 3 Comments
Point 1: Please include the explanation for choosing Qatar under results.
Response 1: A sentence is added in line 85-line 88,
“In the present study, Malaysia and Qatar were chosen because to compare if there were any differences in term of multiple resistant index (MAR), plasmid profiles and DNA heterogeneities between two distantly geographical locations of V. vulnificus isolates”.
Point 2: English language still needs improvement. Please use a software to ensure correct usage.
Response 2: The English language has been improved and this will give better understanding of the manuscript.